# Effect of Arbuscular Mycorrhizal Fungi on Nitrogen and Phosphorus Uptake Efficiency and Crop Productivity of Two-Rowed Barley under Different Crop Production Systems

**DOI:** 10.3390/plants12091908

**Published:** 2023-05-08

**Authors:** Dimitrios Beslemes, Evangelia Tigka, Ioannis Roussis, Ioanna Kakabouki, Antonios Mavroeidis, Dimitrios Vlachostergios

**Affiliations:** 1Institute of Industrial and Forage Crops, Hellenic Agricultural Organization Demeter, 41335 Larissa, Greece; evitiga@yahoo.gr; 2Laboratory of Agronomy, Department of Crop Science, Agricultural University of Athens, 11855 Athens, Greece; iroussis01@gmail.com (I.R.); i.kakabouki@gmail.com (I.K.); antoniosmauroeidis@gmail.com (A.M.)

**Keywords:** AMF root colonization, *Hordeum vulgare* subsp. *distichum* L., nitrogen harvest index (NHI), nitrogen utilization efficiency (NUtE), seed yield, phosphorus uptake

## Abstract

Arbuscular Mycorrhizal Fungi (AMF) constitute a ubiquitous group of soil microorganisms, affecting plant and soil microorganism growth. Various crop management practices can have a significant impact on the AM association. This study investigated the AMF inoculation contribution on growth and productivity of two-rowed barley crop by identifying the underlying mechanisms both in conventional and organic cropping systems. A two-year field trial was set up as a split-plot design with 2 main plots [AMF inoculation: with (AMF+) and without (AMF−)] and five sub-plots (fertilization regimes: untreated, 100% recommended dose of fertilizer in organic and inorganic form, and 60% recommended dose of fertilizer in organic and inorganic form) in three replications. According to the results, AMF+ plants presented higher plant height and leaf area index (LAI), resulting in increased biomass and, as a result, higher seed yield. With regard to the quality traits, including the nitrogen and phosphorus uptake and their utilization indices, the AMF inoculated plants showed higher values. Furthermore, the level of fertilization, particularly in an inorganic form, adversely affected AMF root colonization. Consequently, it was concluded that substitution of inorganic inputs by organic, as well as inputs reduction, when combined with AMF inoculation, can produce excellent results, thus making barley crop cultivation sustainable in Mediterranean climates.

## 1. Introduction

Arbuscular Mycorrhizal Fungi (AMF) constitute a ubiquitous group of soil microorganisms, affecting plant and soil microorganism growth [1,2]. Since mycorrhizal fungi form mutualistic associations with the roots of most terrestrial plants, these symbiotic relationships are common in nature, forming an inherent part of ecosystem functionality [3]. Various types of fungi are responsible for generating these associations with the AMF of the phylum Glomeromycota, forming mutually beneficial relationships with the roots of over 80% of land plants and many agriculturally important crops; however, most species of the Brassicaceae and Chenopodiaceae families are notable exceptions. [4,5].

Various crop management practices can have a significant impact on the AM association. It occurs both directly by causing physical damage or death to AMF, as well as indirectly by creating conditions that can be either beneficial or detrimental to AMF [5,6,7]. Farming practices generally adversely affect the AM relationship, and agricultural soil is depleted of AMF, especially in terms of the number of species [5,7]. Intensive agricultural production, for example, has resulted in the use of phosphorus (P) fertilizers that far exceed crop requirements, resulting in in the accumulation of overall and, in some instances, readily accessible P in the soil [8,9]. Consequently, crops have a reduced dependence on AM association and present lower root colonization and densities of settled propagules [10].

Organic agricultural systems, on the other hand, are heavily dependent on biological processes for nutrient supply due to the exclusion of soluble mineral fertilizers and the limited use of biocides. Due to the assumption that they will compensate for a reduction in the use of P fertilizers, AMF are commonly regarded as playing an important role in this process [10,11]. Despite this, it remains unclear whether AMF actually contributes to the proper functioning of organic agroecosystems, exceptionally in terms of the growth and productivity of the crops. In view of the potential benefits of AM symbiosis, there is an increasing interest in evaluating its effectiveness in specific plant-producing systems and incorporating it into production procedures when possible. In recent years, there has been growing evidence that indigenous and/or introduced AM fungi can contribute to the improved nutrition of annual crops, including barley, bread and durum wheat, sorghum, legumes and other food crops, which are often significantly dependent on AMF and adequate management of AMF has been suggested as one part of the solution to the problem of food security [12,13,14]. In order to apply mycorrhizal technology in agriculture, it is essential to investigate the ecological aspects associated with AM infection and the selection of suitable fungal strains [15,16,17].

A variety of AMF species have been shown to produce different growth responses in the host plant, varying from substantially positive to substantially detrimental [5,16,18]. There is a challenge in selecting the most suitable species for AMF inoculum since the most successful AMF species vary from plant to plant. In this regard, it is important to consider whether the main objective is nutrient uptake, resistance to diseases, or better water distribution. It is possible that the failure to select the appropriate AMF/host/inoculation method is responsible for the failure of several inoculants used to date to produce a positive result, despite the high degree of colonization. It should be noted that, whenever an efficient combination of an AMF, a host, and an inoculant is developed, the problem of vying with the natural soil AMF persists [12,19].

The current research study investigated the contribution of arbuscular mycorrhizal fungi (AMF) inoculation and field AMF communities to two-rowed barley growth and productivity by identifying the fundamental procedures both in conventional and organic crop production systems. As well, this research examined if these crop production systems may affect the colonization of roots by arbuscular mycorrhizal fungi, regardless of whether they are indigenous or introduced through inoculation.

## 2. Results

Based on the two-year data analysis (Table 1), the arbuscular mycorrhizal fungi (AMF) inoculation × fertilization interaction had a substantial impact on plant height, leaf area index (LAI), seed yield, seed phosphorus (P) uptake, total plant P uptake, nitrogen utilization efficiency (NUtE), phosphorus utilization efficiency (PUtE), and AMF inoculation. The main effect of AMF inoculation was significant on all other measures of growth, productivity, and nutrient uptake and utilization efficiency of barley crops except harvest index (HI), nitrogen harvest index (NHI), and NUtE. All traits, except HI, were significantly affected by fertilizers’ application levels and types. Furthermore, the main effect of the year was found to be statistically significant on plant height, biomass yield, seed yield, seed P uptake, as well as total plant P uptake (Table 1).

### 2.1. Growth and Productivity of Two-Rowed Barley

According to Table 2, the plant height of two-rowed barley was significantly influenced by AMF inoculation and showed a similar trend during the two-year study, with the highest values presented in AMF-inoculated (AMF+) plants. Specifically, in the first (2015) experimental year, the plant height of AMF-inoculated (AMF+) plants was greater than that of non-AMF-inoculated (AMF−) plants by 3.98%; whereas, during the second (2016) experimental year, the plant height of AMF+ plants were 4.09% higher than that of AMF− plants. Similarly, the effect of different types and levels of fertilization was also found statistically significant during the experimental periods with the plants receiving full dose of organic or inorganic fertilizer, achieving higher heights, whereas unfertilized plants recorded the lowest values (Table 2).

As presented in Table 1, the leaf area index (LAI) was significantly influenced by both AMF-inoculation and fertilization. Concerning the AMF inoculation effect, the values of LAI in AMF+ plots (3.53 and 3.63 m^2^ m^−2^ in the first and second growing period, respectively) were substantially higher than in AMF− plots (3.28 and 3.35 m^2^ m^−2^ in the first and second cropping seasons, respectively) (Table 2). In regard to the fertilization effect, this had also a significant influence on LAI, with the highest values (4.39 and 4.52 m^2^ m^−2^ in the first and second cropping seasons, respectively) observed in full recommended dose of fertilizer in organic form (100% Org.) in conjunction with AMF+ treatment.

The results of the current study indicated that total biomass yield was significantly affected by AMF inoculation, with the greatest values (7.14 and 7.40 tn ha^−1^ in the first and second cropping seasons, respectively) observed in AMF+ treatments (Table 2). In the same manner, fertilization presented significant impact on the biomass yield, with the highest values observed in full recommended dose of fertilizer in organic form (AMF+: 7.66 and 7.95 tn ha^−1^ in the first and second cropping seasons, respectively; AMF−: 7.38 and 7.66 tn ha^−1^ for the corresponding years), followed by full recommended dose of fertilizer in inorganic form (AMF+: 7.65 and 7.92 tn ha^−1^ in the first and second cropping seasons, respectively; AMF−: 7.34 and 7.61 tn ha^−1^ for the corresponding years).

Concerning seed yield, there were significant differences among AMF-inoculated and non-inoculated plots during the cropping seasons (Table 2), with the highest yields obtained from AMF+ plots, with the values being 4.07 and 4.22 tn ha^−1^ in the first and second cropping seasons, respectively. In regards to the effect of different types and levels of fertilization, the highest seed yields, averaged over AMF treatments, were obtained from plots fertilized with full recommended dose of fertilizer in organic form (100% Org: 4.30 and 4.47 tn ha^−1^ in 2015 and 2016, respectively) and inorganic form (100% Inorg: 4.23 and 4.38 tn ha^−1^ for the respective experimental seasons), while the lowest values (3.33 and 3.46 tn ha^−1^ for the corresponding experimental seasons) were obtained from untreated (control) plot.

As for harvest index (HI), there were no significant differences between AMF inoculation treatments; however, the plants of AMF− plots presented slightly higher value of this measure (58.16% and 58.22% in 2015 and 2016, respectively) than those of AMF+ plots (57.08% and 57.12% for the respective experimental seasons). In the same way, the effect of different types and levels of fertilization was not found to present substantial impact during the two-year study; although slightly greater values (59.72% and 59.89% in 2015 and 2016, respectively) in the case of untreated (unfertilized and AMF−) plots.

### 2.2. Nitrogen (N) and Phosphorus (P) Content, Uptake and Use Efficiency of Two-Rowed Barley

The AMF infection and fertilization effects on the aerial biomass nitrogen (N) uptake of barley are shown in Table 3. In the AMF-inoculated (AMF+) plants, the values of biomass N uptake were significantly greater (17.12 and 17.32 kg N ha^−1^ in the first and the second growing period, respectively) than those of non-AMF-inoculated (AMF−) plants (14.70 and 15.06 kg N ha^−1^ for the corresponding experimental seasons). As for the fertilization effect, during the two-year study, the highest values (23.79 and 25.16 kg N ha^−1^ in 2015 and 2016, respectively) of biomass N uptake were obtained from the fertilization plots treated with full recommended dose of fertilizer in organic form (100% Org.) in conjunction with AMF+ treatment.

Taking into account the combined analysis of variance (Table 1), the seed N uptake was substantially affected by both AMF infection and fertilization during the current two-year research. In regard to AMF inoculation, there were significant differences among AMF inoculated and non-inoculated plots during the experimental periods (Table 1), with the highest values (69.09 and 71.70 kg N ha^−1^ in the first and the second growing period, respectively) obtained from AMF+ plots. Averaged over AMF inoculation treatments, the highest seed N uptake values were found in plots fertilized with full recommended dose of fertilizer in inorganic (100% Inorg: 74.86 and 77.58 kg N ha^−1^ in in the first and the second growing period, respectively) and organic form (100% Org: 74.07 and 76.96 kg N ha^−1^ for the respective experimental seasons).

Total plant (aerial biomass + seeds) N uptake was significantly affected by AMF infection and different types and levels of fertilization. As regards AMF-infection, the greatest values (86.22 and 89.02 kg N ha^−1^ in the first and the second growing period, respectively) were observed when plants infected with AMF. In the same manner, the different fertilization regimes effect was found to present significant differences during the two-year study with greater values (100.64 and 105.07 kg N ha^−1^ in the first and the second cropping period, respectively) found in the case of full recommended dose of fertilizer in organic form (100% Org.) in conjunction with AMF+ treatment.

The results of the current experiment indicated that nitrogen harvest index (NHI) was not influenced by AMF inoculation during the cropping seasons; however, the non-inoculated plants (AMF−) presented slightly higher of NHI index (81.91% and 82.11% in the first and the second growing period, respectively) than those of AMF-infected (AMF+) plants (80.45% and 80.91% for the corresponding experimental seasons). As for the different fertilization regimes, these had a substantial effect according to the combined analysis of variance (Table 1), and the highest NHI values, averaged over growing years and AMF inoculation treatments, were achieved in 60% recommended dose of fertilizer in inorganic form (83.52%) and unfertilized (control: 81.96%), while the lowest value (79.00%) was obtained in plots of full recommended dose of fertilizer in organic form.

Phosphorus (P) uptake in aerial biomass was significantly higher in the AMF-inoculated plants (AMF+) than in non-inoculated plants (AMF−) during the cropping periods, with the values in AMF+ plants being 3.87 and 4.06 kg P ha^−1^ in in the first and second cropping seasons, respectively (Table 3). In regards to different fertilization regimes, these had a significant effect, and the greatest values (4.67 and 4.83 kg P ha^−1^ for the respective experimental seasons) of biomass P uptake were obtained from the fertilization plots treated with 60% recommended dose of fertilizer in organic form (60% Org.) in conjunction with AMF+ treatment.

According to the results of the study, P uptake in seeds was influenced by both AMF infection and different levels and types of fertilization during the growing seasons (Table 1). Concerning AMF inoculation effect, the seed P uptake recorded in AMF-inoculated (AMF+) plots (8.40 and 8.85 kg P ha^−1^ in the first and the second growing period, respectively) were higher than in non-AMF-infected (AMF−) treatments (8.11 and 8.37 kg P ha^−1^ for the corresponding experimental seasons) (Table 3). Regarding the fertilization regimes effect, the highest values (9.63 and 10.16 kg P ha^−1^ for the respective experimental seasons) of seed P uptake were recorded in the fertilization plots treated with full recommended dose of fertilizer in organic form in conjunction with AMF+ treatment.

Total plant P uptake was also significantly influenced by AMF inoculation and the different types and levels of fertilization treatments. Averaged over experimental years and fertilization regimes, the highest value (12.59 kg P ha^−1^) was recorded when plants subjected to AMF inoculation (Table 3). Concerning the effect of different fertilization regimes, the highest total plant P uptake values, averaged over planting years and AMF inoculation, were recorded in 100% of recommended dose in organic (13.11 kg P ha^−1^) and inorganic form (13.09 kg P ha^−1^), while the lowest value (9.63 kg P ha^−1^) was obtained in the unfertilized (control) plots.

Phosphorus harvest index (PHI) was substantially affected by AMF infection and different fertilization regimes during the two-year experiment. As regards AMF infection, there were substantial differences among AMF infected and non-AMF-infected plots during the growing seasons, with the highest values (73.03% and 73.15% in the first and the second growing period, respectively) obtained from AMF− plants. Averaged over AMF inoculation treatments, the highest PHI values were found in unfertilized plots in conjunction with AMF− treatment (74.43% and 75.10% in the first and the second growing period, respectively).

The utilization efficiency of nitrogen element (N utilization efficiency, NUtE; kg kg^−1^) was expressed as the ratio (production in Nx − production in N0) to (N absorbed in Nx − N absorbed in No), as presented in Figure 1A diagrammatically. NUtE recorded a decrease in the non-AMF-infected plants from 35.02 kg kg^−1^ to 23.12 kg kg^−1^. Moreover, for AMF infected plants, a linear association is found among nitrogen which was absorbed by plants and the final yields in seed. In a similar manner to NUtE, the efficiency of phosphorus utilization (PUtE) was defined as the ratio of (production at Px − production at P0) to (P absorbed at Px − P absorbed at P0), and is illustrated diagrammatically from the curve slopes in Figure 1B. The productivities were close to the maximum, as with the increasing application of phosphorus (in both fertilization types) the PUtE strayed from linear regression, signifying high phosphorus concentrations in plant tissues (luxuriant growth). PUtE values for barley ranged from 270 kg kg^−1^ to 240 kg kg^−1^ for AMF infection or not, respectively.

### 2.3. Arbuscular Mycorrhizal Fungi (AMF) Colonization and Weighted Mycorrhizal Dependency (WMD) of Two-Rowed Barley

The AMF infection and fertilization effects on AMF colonization are shown in the diagrams of the following figure (Figure 2). In AMF+ plants, the values of AMF colonization were significantly higher (53.69% and 54.71% in the first and second crop growing period, respectively) in comparison to those of AMF− plants (48.56% and 48.68% for the corresponding experimental seasons). Furthermore, the mean valued of AMF colonization presented significant evidence of the effect of different types and levels of fertilization. Averaged over AMF inoculation, the greatest values were recorded in unfertilized (control) plots (55.61% and 55.68% in 2015 and 2016, respectively), followed by 60% recommended dose of fertilizer in organic form (52.52% and 54.44% for the respective experimental seasons), and full recommended dose of fertilizer in organic form (53.15% and 52.14% for the corresponding experimental seasons).

As shown in Figure 3, the weighted mycorrhizal dependency (WMD) of two-rowed barley plants was significantly affected by the different types and levels of fertilization regimes. Specifically, the greatest WMD values (25.53% and 24.57% in 2015 and 2016, respectively), were recorded in unfertilized (control) plots, whereas the lowest values (6.59% and 3.67% for the respective experimental seasons), were obtained in the plots fertilized with 60% recommended dose of fertilizer in organic form.

## 3. Discussion

During the two-year field experiment, the combined analysis of variability revealed that the plant height of two-rowed barley crop was affected by AMF inoculation, and the highest two-year average value was obtained from the AMF-inoculated plants, with the value being 4.04% higher than the non-inoculated plants. This could be owing to the positive response of AMF infection, which is capable of increasing the uptake of nutrients and water in the plant body, resulting in increased plant height [5,20]. 

AMF and plant roots have a mutually beneficial relationship, which is based on the exchange of nutrients among the fungi and host plants [21]. As a reciprocal exchange, host plants provide AMF with carbon (photosynthates and fatty acids), whereas AMF supply hosts with essential minerals (phosphorus (P) and nitrogen (N)) [21,22]. Additionally, AM symbiosis enhances plant growth, productivity, and resistance to biotic and abiotic stresses [23,24,25]. In general, the development of the AM symbiosis can be divided into four phases [26]. In the first phase, the two symbionts cross-talk using diffusible signal molecules; in particular, fungi typically exude lipo-chitooligosaccharides, whereas plants produce strigolactones. The second phase includes the physical contact among root surfaces and hyphae, followed by the formation of hyphopodium on the root surfaces. The third phase involves the penetration and proliferation of fungal hyphae into the apoplast of the cortex, altering their typical appearance. In the last phase, arbuscules are formed by fungal hyphae penetrating and multiplying in the inner cortex, resulting in colonization of the surface [26]. Both symbionts exchange nutrients through arbuscules, and the formation of these arbuscules causes de novo synthesis of the peri-arbuscular membrane surrounding the cytoplasm [27]. An interaction between translocation factors and phytohormones (auxin, gibberellin, abscisic acid, strigolactones and ethylene) regulates the symbiosis signaling pathway and symbiont interactions [26,28]. Through the formation of arbuscules, the expression patterns of many genes in AM roots are altered, which results in a variation in primary and secondary metabolites as well as improved production [29]. In addition, there is evidence that AM-induced molecular mechanisms depend on species, genotypes, types of AMF, and growth conditions [30].

In terms of fertilization, it was found that plants treated with full N and P fertilization, either organically or inorganically, reached significantly higher final heights for both AMF treatments (Table 2), whereas the unfertilized plants reached only minor heights. As a result of nitrogen fertilization, plants during the vegetative phase were likely to grow at a faster rate [31]. When the nitrogen level of barley plants was increased, the plant height increased primarily because nitrogen stimulated metabolic activity, resulting in an increase in metabolites amount, which in turn resulted in internodes’ elongation and increased plant height by enhancing the plant availability for nitrogen [32]. In light of the relatively high height of the barley plant, a significant high concentration of photosynthetic products was found, and the plant height correlated positively with total biomass yield (*r* = 0.788, *p* < 0.001; Figure 4).

The current study found that the leaf area index (LAI) of barley was significantly affected by both AMF inoculation and fertilization regimes. As regards the AMF inoculation effect, the two-year mean value of LAI found in AMF+ plants (3.58 m^2^ m^−2^) were greater than that of AMF− plants (3.32 m^2^ m^−2^). A number of plant species have demonstrated the beneficial effects of mycorrhizal fungi on plant growth, including faba bean [33], maize [34], tomato [35], and upland cotton [36]. In addition, fertilization type and level had a substantial impact on LAI, with the greatest two-year mean value (4.46 m^2^ m^−2^) obtained from plots fertilized with the full recommended dose of fertilizer in organic form in conjunction with AMF+ treatment, substantiating that replacing inorganic inputs with organic ones coupled with AMF inoculation would not lead to an increase in the risk of an open canopy (and thus a substantial loss of productivity) and is appropriate to be considered in a sincere manner as a sustainable agronomic practice [35,37].

In regard to the total plant biomass, the highest values obtained from AMF+ plants with the 2-year mean value being 8.02% greater in comparison to AMF− plants. Several other researchers [38,39] have shown that plants with AMF can increase aerial biomass dry weight as well as crop grain yield, while accumulating nitrogen and phosphorus. It is most likely that the AMF inoculation was responsible for the increased biomass accumulation. Additionally, the type of applicated fertilizer (organic or inorganic) had a substantial impact with both types of fertilizers producing high levels of biomass when applied at full recommended dose (Table 2). With the enhanced growth in the aerial plant parts, and specifically the leaf area, coupled with the higher levels of nitrogen in full dose of organic and inorganic treatments, higher rates of photosynthesis were achieved, which resulted in the highest biomass yields. Based on the significant positive correlation coefficient between total biomass yield and leaf area index of the barley crop (*r* = 0.505, *p* < 0.001; Figure 4), it can be concluded that this relationship exists. As soluble protein, carboxylase activity, and chlorophyll and total nitrogen content increase, the rate of photosynthesis increases as well. As a result, the entire seed and plant are supplied with photosynthetic products [40,41].

The combined analysis of variance shown that the seed yield of two-rowed barley was substantially affected by AMF inoculation treatment. In particular, the highest two-year mean value was obtained from AMF+ plots, with the value being 9.21% higher than the AMF− treatment. The results of the current research indicated that AMF increased barley seed yield in accordance with several previous studies [42,43]. In general, global meta-analyses indicate that AMF inoculation increased crop seed yield by approximately 20% [44,45]. In response to different fertilization regimes, both types of fertilization (organic or inorganic) produced significant amounts of seeds when used at 100% of the advised dose (Table 2). Based on the abovementioned results, it was confirmed that an increase in nitrogen rates positively and linearly affected seed yield of the barley crop [46]. Furthermore, in the present research, the leaf area index (LAI) was linearly related to seed production (*r* = 0.682, *p* < 0.001; Figure 4). As a result of increased photosynthesis, an increase in photosynthetic products was produced, which resulted in an increase in biomass and seed yield [32,41,47].

Concerning harvest index (HI), neither the AMF inoculation nor the various fertilization regimes had an impact on this trait. At this point, it is important to note that the AMF− plants and those that not received fertilization displayed marginally higher values (Table 2). Generally, the values of HI were ranged between 56.00% and 59.89%. The findings are consistent with those reported by various researchers who studied the development of barley crop under Mediterranean conditions [48,49].

Aerial biomass nitrogen (N) uptake of barley was affected by both experimental factors (Table 1). In particular, the AMF+ plants with a two-year average value of 17.22 kg N ha^−1^, was substantially superior to that of AMF− plants with an average value of 14.88 kg N ha^−1^. As for different fertilization regimes, it was found that plants treated with full nitrogen (N), especially in organic form, in combination with AMF+ treatment reached significantly higher values (Table 3). The amount of nitrogen which is absorbed in biomass is a function of biomass multiplied by the amount of N present in the plant tissues, and the variableness of the values of this trait is dependent upon how closely they are related. Under low N inputs, a strong negative association between productivity and the proportion of N in the plant tissues (solution effect) is presented, thereby reducing the variableness of absorbed N regulated by genetic factors [50]. Unlike low N inputs, high N inputs are associated with considerable variability in absorbed nitrogen, which does not negatively correlate with production and tissue N content [40]. N uptake in plants is generally improved by increasing the amount of N available for uptake [51].

AMF inoculation had a significant influence on seed nitrogen (N) uptake of barley crop throughout the two-year trial. Specifically, seed N uptake was greater in AMF+ plants with the average two-year value being 4.40% higher than in AMF− plants. In terms of different types and levels of fertilization, there was a substantial effect with the two-year average value of seed N uptake being significantly higher in the plots that had received the full nitrogen (N) fertilization, either organically or inorganically (Table 3). Increasing attention is being paid to organic N fertilization as a result of the development of more sustainable agricultural techniques [47]. Agricultural soils fertilized with organic N may have reduced the potential for mycorrhizal inoculum [52], increased the severity of pathogens [16,53], and increased greenhouse gas emissions [54]. Combining organic N fertilization with AMF inoculation could provide a solution to some of these issues and improve the uptake of N fertilizer into crop plants, which is now restricted to 40–60% in crop plants that receive inorganic N fertilization [16,55,56]. Moreover, the increased amount of available N in the plant caused an increase in the accumulation of this element in the seeds, indicating that administered N is one of the most important factors in increasing the content of total N in seeds. These results are similar to those of other researchers who reported that N concentrations in plant biomass and then in seeds increased with a greater amount of available N [49,57,58].

Concerning the absorption of nitrogen (N) in total aerial dry matter (Total plant N uptake), the combined analysis revealed that both experimental factors significantly affected the present characteristic. As for AMF inoculation effect, the two-year mean value of total plant N uptake in AMF+ plants (87.62 kg N ha^−1^) were greater than that of AMF− plants (82.32 kg N ha^−1^). As for different fertilization regimes, both types of fertilization (organic or inorganic), and especially the organic ones, produced significant amounts of seeds when used at the full advised dose (Table 3). The transmission of N by AMF to their associated hosts has been reported [59,60]; however, many questions remain regarding the ecological implications [61]. There is still no consensus on the precise process of N transfer, in addition to the quantities of N that are transmitted through the AMF in relation to the plant’s requirements for N [61]. Even though data from root organ culture studies show that the AMF pathway is capable of obtaining up to 50% of root nitrogen [62], implication of much about all plant nutrient dynamics may be unsafe. In light of the growth conditions used, source-sink interactions are most likely impractical [61]. It has been demonstrated that the use of AMF can provide up to 15–20% of the plant’s total N uptake when it is added in the form of organic matter patches [63,64]. Due to its role as a component of chlorophyll, nitrogen plays an essential role in photosynthesis. Photosynthetic resources transported to the roots stimulate the activity of soil microorganisms, for instance AMF and plant growth promoting bacteria (PGPB) as well [65]. Furthermore, as determined and discussed by Plaza-Bonilla [49], Dordas [57], and Bulman and Smith, [66], the response trends for N uptake into seeds and total above-ground biomass at different levels of fertilization are similar to crop yield and total crop dry weight, respectively. This is supported by the considerable positive associations between seed N uptake and seed yield (*r* = 0.888, *p* < 0.001) and total plant N uptake and crop above-ground biomass (*r* = 0.895, *p* < 0.001) (Figure 4).

The nitrogen harvest index (NHI) is a critical metric for determining how efficiently absorbed N is transported from plant vegetative parts to seeds. This measure is critical for tracking N dispersion in crop plants since it reveals how well absorbed nitrogen was used for seed formation [67]. High NHI readings indicate that nitrogen in seeds is spread more uniformly [66]. In the current study, the NHI index was only influenced by different fertilization regimes the highest values of this index were achieved in 60% of recommended dose in inorganic form and unfertilized (Table 3). Furthermore, the influence of the investigated treatments on the NHI were proportionate to the harvest index (HI). This is further supported by the positive and significant relationship between these assessed characteristics (*r* = 0.567, *p* < 0.001; Figure 4).

Throughout the two-year trial, the aerial biomass phosphorus (P) uptake of barley was statistically affected by AMF inoculation, with the two-year mean value of AMF+ plants (3.05 kg P ha^−1^) being greater than that of AMF− plants (3.32 kg P ha^−1^). In terms of different fertilization regimes, the highest values of this trait were found in plots treated with 60% recommended dose of fertilizer in organic form (60% Org.) in conjunction with AMF+ treatment (Table 3). In comparison with the control, aerial biomass P uptake was enhanced by AMF inoculation. Similarly, Najafi et al. [68] and Masahari et al. [43] demonstrated that microbial symbiosis promotes root growth, water absorption, and nutrient utilization in barley. There is a positive effect of AMF on the growth of hair roots. Due to this, the mycelium of the fungi grows longitudinally, allowing it to penetrate deeper layers of soil, thereby improving nutrient availability [69,70]. As a consequence of AMF, inorganic minerals and carbon and P-containing molecules are exchanged, giving host plants a greater level of vitality [71]. Therefore, they can significantly increase both root and shoot P levels. As a result of mycorrhizal associations, phosphorus is further supplied to the infected roots of host plants in conditions of low P availability [5]. AMF can optimize P solubilization, increase nutrient levels, and mineralize organic phosphates [43,72,73]. Accordingly, AMFs are evidently capable of improving the uptake of inorganic nutrients in most plants, especially phosphate [74]. In addition, organic fertilizers may also indirectly positively impact mycorrhiza-mediated nutrient uptake due to the growth of extraradical hyphae promoted by greater soil organic matter [43,75]. Rather than fungus-specific P affinity, mycorrhizal plants seem to rely on extraradical hyphal lengths to determine P uptake [5,76].

The results of the absorption of phosphorus (N) in seeds (seed P uptake) and in total aerial dry matter (total plant P uptake), are presented in Table 3. At different AMF colonization and fertilization types and levels, seed P uptake and total aerial biomass yield exhibited similar trends, as determined and clearly described by Masahari et al. [43] and Heydari and Maleki [73]. The present experiment confirms this by showing significant positive correlations between seed P uptake with seed yield (*r* = 0.913, *p* < 0.001) and total aerial biomass yield (*r* = 0.878, *p* < 0.001) (Figure 4). As for total plant P uptake, this trait’s trends, like seed P uptake, are consistent with seed and aerial biomass yield [43], as evidenced by the strong and positive correlations of total plant P uptake with seed yield (*r* = 0.903, *p* < 0.001) and aerial biomass yield (*r* = 0.933, *p* < 0.001) (Figure 4).

Another important parameter is the phosphorus harvest index (PHI), which is determined by dividing seed phosphorus with the total amount absorbed by the plant. In the current research, as regards the AMF inoculation effect, the two-year mean value of PHI observed in AMF− plants (73.09%) was greater than that of AMF+ plants (68.58%). Moreover, fertilization type and level had a substantial effect on PHI, with the greatest values obtained from unfertilized plots in combination with AMF− treatment (Table 3). PHI impacts the nutritional quality of the seed, but research attempting to determine the genetic basis of seed composition has extrapolated a relationship between yield and phosphate content appears to be inversely proportional, inhibiting the effects of cultivation techniques. The variation recorded for PHI indicates that phosphorus accumulation and mobilization, although largely controlled by genotype, is also affected by factors that could affect the range of transportation of pre-flowering assimilated components and phosphorus to the seed. The reduction in PHI may also be explained by the fact that AMF inoculation significantly increases phosphorus uptake on leaves and stems, as confirmed by analyses revealing an increase in phosphorus uptake on leaves and stems. As a result of reducing the total biomass, or total seed, plants are able to grow in an environment without excessive amounts of phosphorus. In addition, it should be noted that phosphorus recovery fractions increased only modestly as a result of the high AMF native population in the field where the experiment took place [43,73,77].

The AMF inoculation effect on nitrogen utilization efficiency (NUtE) of barley is shown in Figure 1A. AMF+ plants showed a linear connection between nitrogen absorbed and seed production, suggesting that maximum potential production was not achieved under conditions of deprivation owing to low nitrogen concentrations in plant tissues [78]. Conversely, in plots without the inoculation of AMF, productivity was near maximum, as NUtE deviated from linearity as nitrogen application increased (organic or inorganic), indicating higher nitrogen concentrations in plant tissues (luxurious growth). As a result of increased nitrogen uptake in the shoots and seeds aforementioned, NUtE values for a given production level are lower than the maximum value corresponding to the maximum nitrogen uptake required for optimum seed yield [49,57,79].

As for the AMF inoculation influence on phosphorus utilization efficiency (PUtE), this is presented in Figure 1B. In the current study, seed productivities for AMF and non-AMF inoculated plants were near to maximum, as with rising P (in organic or inorganic form) application, PUtE diverged from linearity, indicating high P concentrations in plant tissues (luxuriant growth). Phosphorus absorption upwards of the quantities necessary to ascertain the optimum yield in seed results in lower PUtE values in comparison to the corresponding maximum value for the specific production level is connected to the increased P concentration in shoots and seed discussed above. [43,68].

As demonstrated in Figure 2A, mycorrhizal settlement was more prevalent in plants in which AMF inoculation was used. This demonstrates the effectiveness of the vaccine on the one hand and the efficacious impact of soil enrichment on the population as a function of AMF number on the other. Furthermore, it is evident that mycorrhizal colonization was high in the non-AMF-inoculated plots, indicating that a suitable, robust, native mycorrhizal population exists in the experimental field as a consequence of many years of appropriate crop management practices, including limiting use of pesticides and fertilizers, crop rotation, and previously legume crops cultivation. Ultimately, there is a strong negative association between the percentage of settlement and the fertilizer inputs quantities, particularly for inorganic inputs (Figure 2B). The negative association is weaker for organic fertilization. This could be owing to the gradual release of nutrients in organic fertilizers, which increases the reliance of nutrient uptake on fungus cooperation, and vice versa [43,76].

The negative connection between AMF root colonization and fertilization levels is additionally expressed in the degree of mycorrhizal dependence of barley cultivation, according to Figure 3, which depicts the crop’s weighted mycorrhizal dependence (WMD), indicating the true variations that occur once the soil’s inherent presence of mycorrhizae is considered. With an increase in fertilization level, especially in the organic fertilization, there is a reduction in mycorrhizal dependence, but there is a much smaller decrease in dependence with respect to inorganic fertilization.

AMF can fulfill a vital function in plant P nutrition by improving total absorption and, in some situations, the effectiveness of P utilization, which can benefit plant growth and development and enhance the crop production [43,75,76]. Once AMF colonization is interrupted, P absorption, growth, and development, as well as, in some situations, productivity, could suffer. In quite a while, there is a number of plants failing to respond to native AMF settlement, owing to excessive P concentrations in plants in the soil [5,80,81]. The AMF colonization of roots is typically reduced under these conditions [65,82], whereas significant settlement at high concentrations of P in soil results in the suppression of plant growth [72]. It is possible for plants to fail to respond to native AMF or to AMF vaccines in soils with low levels of phosphorus, despite any increase in AMF colonization, for reasons that are unknown [5,81,82].

## 4. Materials and Methods

### 4.1. Site Description and Experimental Design

The experiment on two-rowed barley (*Hordeum vulgare* subsp. *distichum* L. cv. Triptolemos) crop was conducted over a period of two growing seasons (2015–2016) in the experimental field of the Institute of Industrial and Forage Crops (Latitude: 39°30′ N, Longitude: 22°42′ E, Altitude: 77 m above sea level) (Figure 5). The soil was characterized as a clay (51% clay, 23% silt, 26% sand) with soil pH at 7.2 (1:1 H_2_O), organic matter content at 1.6%, total available nitrogen at 900 mg kg^−1^, available potassium at 1.7 cmol kg^−1^, and phosphorus availability at 6.7 mg kg^−1^. The weather data concerning mean temperature and precipitation at the experimental site during the experimental periods were based on observations made by the Network of Agro-Meteorological Stations Horta Srl., and are presented in Figure 6. A Mediterranean climate is generally observed in the study area (Larissa, Thessaly, Greece), where the summers are hot and dry and the winters are cool and humid. Weather data showed that the mean temperature during the cultivation period (from January to June) was 12.7 °C in 2015 and 14.6 °C in 2016, with precipitation amounts of 316.2 mm and 266.2 mm, respectively, during 2015 and 2016.

The experimental area was, in total, 588 m^2^. The trial was set up as a split-plot design with two main plots (AMF+ and AMF−, with and without AMF inoculation, respectively) and five sub-plots [fertilization treatments: untreated (control), 100% and 60% recommended dose of fertilizer in organic form (100% Org and 60% Org, respectively), and 100% and 60% recommended dose of fertilizer in inorganic form (100% Inorg. and 60% Inorg, respectively)] in three replications. Both main plots and sub-plots were relocated in the same field during second year and inoculation was repeated. In the case of AMF inoculation, furrows 25 cm apart were opened to a depth of approximately 5 cm and AM fungi inoculum was evenly distributed at the bottom of the furrows of the AMF-inoculated plots. The inoculum utilized in the current study was a mixture of AM fungi corresponding to the composition of a fungal part of a commercial product Symbivit (Symbiom Ltd., Lanškroun, Czech Republic), which contained six AM fungi species from the phylum Glomeromycota (*Claroideoglomus etunicatum*, *C. claroideum*, *Funneliformis geosporum*, *F. mosseae*, *Rhizoglomus microaggregatum*, and *Rhizophagus intraradices*) in a form of AM colonized root pieces and spores as well as hyphae in fritted clay. According to the manufacturer’s guidelines, the AMF inoculum application rate was 65 kg ha^−1^ Fertilizers application was performed in two potions. The first portion was added at sowing stage as basal dressing with 50 kg N ha^−1^, 40 kg P ha^−1^ and 15 kg K ha^−1^ in all plots for the full dose (100%), and the second one (equal amount of N) was added after the beginning of vegetative phase. The type of inorganic fertilizer was 18-9-6 (+9 S) + 0.2 Zn (Nutrifert^®^ Starter; Hellagrolip S.A., Athens, Greece), whereas the type of organic ones was 10-4-3 (Biofertin^®^; Humofert S.A., Metamorfosi, Greece). The main plot and sub-plot units were 75 m^2^ (15 m × 5 m) and 15 m^2^ (5 m × 3 m), respectively. The soil was prepared two days before sowing by mouldboard ploughing at a depth of 0.25 m. Hand broadcasting of barley seeds was done at a depth of 3–5 cm, with row and intra-row spacing of 15 and 5 cm, respectively. The sowing rate was 170 kg ha^−1^, and seed sowing took place on 22 January 2015 and 26 January 2016. During the two-year trial, no irrigation was used. There was also no evidence of macronutrient deficiency, water stress, or disease on the barley crops. In addition, hand-hoeing was used to manage weeds as necessary and prior to canopy closure.

### 4.2. Sampling, Measurements and Methods

At 100 days after sowing (DAS), a sample of twenty randomly selected plants from each subplot was used to determine plant height and leaf area index (LAI). In order to calculate the LAI index, the plant sample leaves were scanned on a high-resolution scanner (HP Scanjet 200 Flatbed Photo Scanner; Hewlett-Packard Inc., Palo Alto, CA, USA) using ImageJ Ops software (Laboratory for Optical and Computational Instrumentation, University of Wisconsin, Madison, WI, USA) [83]. Consequently, LAI was calculated by dividing the plant-based measurements by the average crop density of each plot. Furthermore, at 130 DAS, twenty plant samples were randomly picked from each sub-plot. The plants, divided into stems, inflorescences, seeds, and leaves, were weighted and then oven-dried for 48 h at 70 °C. To determine the total nitrogen (N) content in the aerial biomass and seeds, the powdered samples were ground to a fine powder and the Kjeldahl method was applied using a Kjeltec 8400 autoanalyzer (Foss Tecator AB, Höganas, Sweden). In addition, at the same time (130 DAS), the determination of phosphorus (P) content took place by selecting twenty plants per sub-plot. In particular, the collected plants were separated into aerial biomass and seeds, dissolved in an HNO_3_H_2_O_2_ solution, and heated under pressure in a CEM MDS 2000 microwave digestor oven (CEM Microwave Technology Ltd., Buckingham, UK). After that, the extract was then analyzed with an iCAP 6500 DUO inductively coupled plasma emission spectrometer (Thermo Fisher Scientific, Waltham, MA, USA). The total plant N and P uptake was calculated as the amount of the corresponding nutrient absorbed in the aboveground (aerial biomass as well as seeds) dry matter at full maturity stage (130 DAS). The N and P harvest indices (NHI and PHI) were estimated by dividing seed nutrient absorption with total plant nutrient uptake, as follows [84]:(1)$$\mathrm{NHI}=\frac{seed\;N\;uptake\;\left(\mathrm{kg}\;N\;{\mathrm{ha}}^{-1}\right)}{total\;plant\;N\;uptake\;\left(\mathrm{kg}\;N\;{\mathrm{ha}}^{-1}\right)}$$
(2)$$\mathrm{PHI}=\frac{seed\;P\;uptake\;\left(\mathrm{kg}\;P\;{\mathrm{ha}}^{-1}\right)}{total\;plant\;P\;uptake\;\left(\mathrm{kg}\;P\;{\mathrm{ha}}^{-1}\right)}$$

The N and P utilization efficiencies (NUtE and PUtE) were derived by dividing seed yield with total plant nutrient uptake, as given by Fageria and Baligar [67] and Ye et al. [84]:(3)$$\mathrm{NUtE}=\frac{seed\;yield\;\left(\mathrm{kg}\;{\mathrm{ha}}^{-1}\right)\;}{total\;plant\;N\;uptake\;\left(\mathrm{kg}\;N\;{\mathrm{ha}}^{-1}\right)}$$
(4)$$\mathrm{PUtE}=\frac{seed\;yield\;\left(\mathrm{kg}\;{\mathrm{ha}}^{-1}\right)\;}{total\;plant\;P\;uptake\;\left(\mathrm{kg}\;P\;{\mathrm{ha}}^{-1}\right)}$$

In addition, at 130 DAS, three root samples from each plot were examined. In particular, a cylindrical auger (25 cm length, 10 cm diameter) was used to collect root samples from the 0–35 cm layer at the midpoint between subsequent plants within a row. In each sample, roots were separated by soaking the sample in 30 mL of a 0.5% solution of sodium hexametaphosphate overnight. Following that, the sample was mixed for 5 min before being rinsed through a 5 mm mesh screen. According to Kormanik and McGraw’s [85] method, the roots kept on the sieves were washed and colored with lactic acid/fuchsin. At a magnification of 30–40×, the percentage of AMF root colonization was assessed microscopically using the gridline-intersection method [86]. Weighted mycorrhizal dependence (WMD) was determined using a modified equation of Plenchette et al. [87] as follows:(5)$$\mathrm{WMD}\;(\%)=\frac{\left(dry\;weight\;of\;inoculated\;plant\times \%\;colonization\right)-\left(dry\;weight\;of\;non-inoculated\;plant\times \%\;colonization\right)\;}{\left(dry\;weight\;of\;inoculated\;plant\times \%\;colonization\right)}\times 100$$

The adjustment integrates the influence of indigenous AMF populations on crop production, since not inoculated plots hosted native AMF populations. Indigenous AMF populations were predicted to colonize barley crops in control plots, implying that these plots’ plants were only non-inoculated and not non-mycorrhizal. The genera Glomus and Gigaspora have been reported to be the dominant AM fungi in the arable lands of Greece [88,89].

The crop was harvested on 4 June 2015 (133 DAS) and 9 June 2016 (135 DAS), once the seeds had attained full maturity (13% seed moisture) Plants originating from the middle subplot area (2 m^2^) were used to calculate the seed yield. The harvest index (HI) is calculated based on the ratio between seed yield and above-ground biomass yield (total weight of plants derived for seed yield).

### 4.3. Statistical Analysis

The analysis of variance of the current study was carried out using the SPSS logistic package (IBM SPSS Statistics 22; IBM Corp., Armonk, NY, USA). The trait data generated by AMF inoculation and fertilization treatments over the 2-year experiment were assessed using a 2 × 2 × 5 factorial design (two experimental years; two AMF-inoculation treatments and five fertilization treatments) set up in a split-plot design with 3 replications. A mixed model was used for the analysis of variance (ANOVA), with years and replications as random effects and AMF-inoculation and fertilization as fixed effects. For the purpose of finding differences between means, the LSD_0.05_ test was used, along with Tukey’s honestly significant difference (Tukey’s HSD) test at the 5% level of significance (*p* = 0.05). In addition, a simple regression analysis was performed to evaluate the relationships between the variables analyzed (*p* = 0.05).

## 5. Conclusions

Based on the findings of this research and their assessment, it is evident that AMF inoculation can positively affect the growth and productivity of two-rowed barley crop, whether it is produced organically or conventionally. There was a significant increase in plant height and leaf area index in plots inoculated with AMF, which resulted in an increase in biomass, as well as a greater overall seed yield, even at lower inputs of N and P. Similar results to those of the productivity results were obtained regarding quality parameters, including nutrient uptake and utilization indexes, with AMF inoculated plants showing higher results. Furthermore, the level of fertilization, particularly in an inorganic form, adversely affected AMF root colonization. Consequently, it was concluded that substitution of inorganic inputs by organic, as well as reduction of inputs, when combined with AMF inoculation, can produce excellent results, thus making barley crop cultivation in organic and low inputs systems sustainable in Mediterranean climates.

## Figures and Tables

**Figure 1 plants-12-01908-f001:**
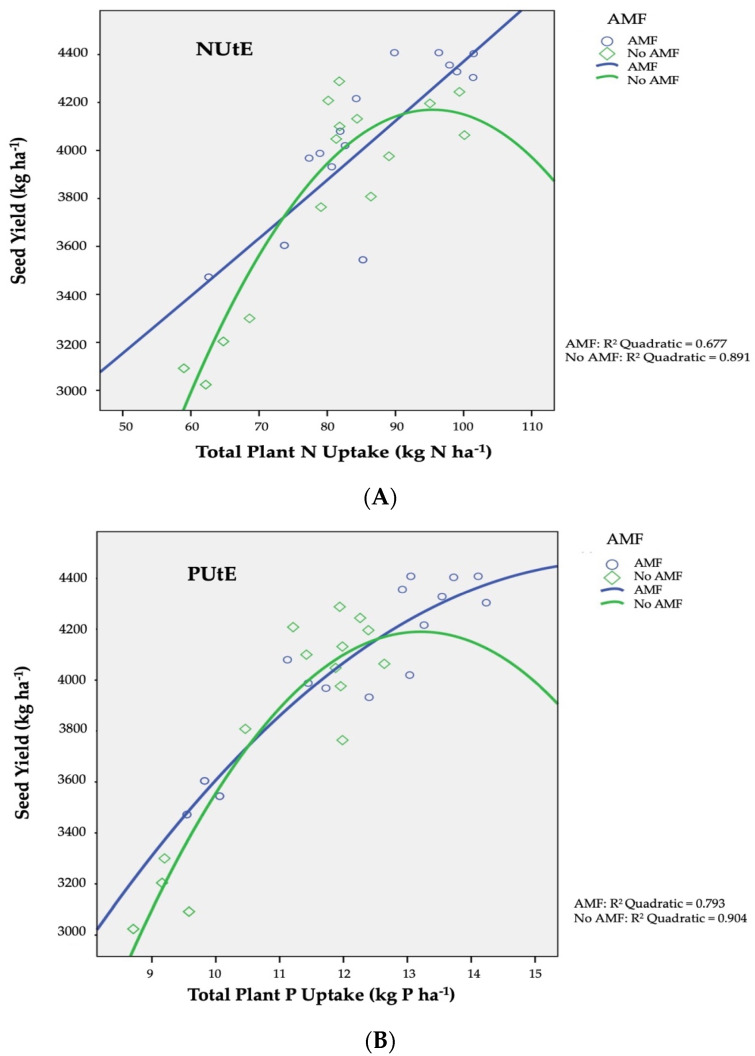
AMF inoculation effect on (**A**) nitrogen utilization efficiency (NUtE) and (**B**) phosphorus utilization efficiency (PUtE) of two-rowed barley crop. AMF inoculation treatments: AMF and No AMF, with and without AMF inoculation, respectively.

**Figure 2 plants-12-01908-f002:**
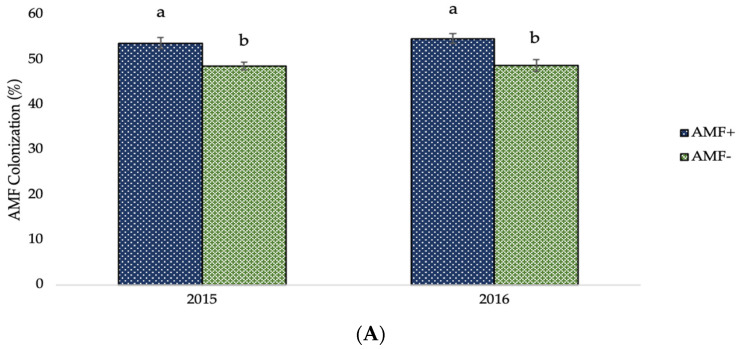
AMF colonization of two-rowed barley crop as affected by (**A**) AMF inoculation and (**B**) fertilization. Error bars represent ± S.E. Different letters above bars represent significant differences at *p* < 0.05 according to Tukey’s HSD test. AMF inoculation treatments: AMF+ and AMF−, with and without AMF inoculation, respectively. Fertilization treatments: 100% and 60% recommended dose of fertilizer in organic form (100% Org and 60% Org, respectively), 100% and 60% recommended dose of fertilizer in inorganic form (100% Inorg. and 60% Inorg, respectively), and untreated (control).

**Figure 3 plants-12-01908-f003:**
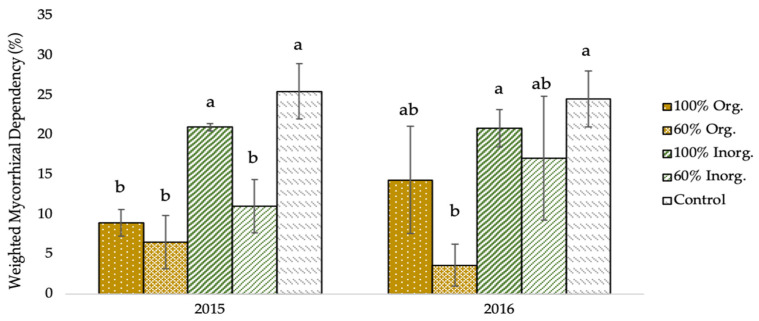
Weighted mycorrhizal dependency (WMD) of two-rowed barley crop as affected by different fertilization type and level. Error bars represent ± S.E. Different letters above bars represent significant differences at *p* < 0.05 according to Tukey’s HSD test. Fertilization treatments: 100% and 60% recommended dose of fertilizer in organic form (100% Org and 60% Org, respectively), 100% and 60% recommended dose of fertilizer in inorganic form (100% Inorg. and 60% Inorg, respectively), and untreated (control).

**Figure 4 plants-12-01908-f004:**
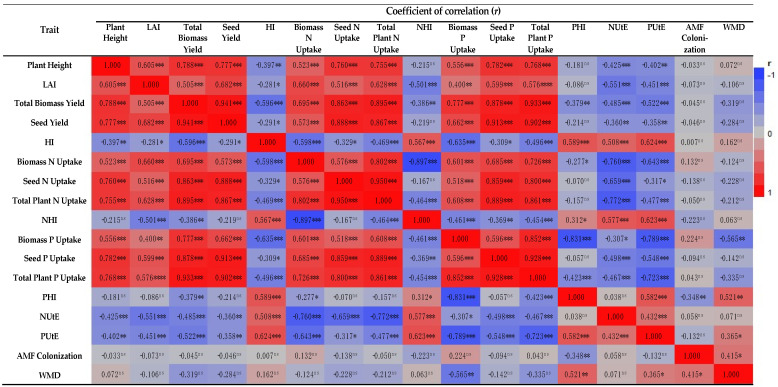
Heatmap of correlation coefficients between evaluated traits. ns: Non-significant; *, **, *** and ****: Significant at the 5%, 1%, 0.1% and 0.01% levels, respectively. LAI: leaf area index, HI: harvest index, N: nitrogen, P: phosphorus, NHI: nitrogen harvest index, PHI: phosphorus harvest index, NUtE: nitrogen utilization efficiency, PUtE: phosphorus utilization efficiency (PUtE), WMD: weighted mycorrhizal dependency.

**Figure 5 plants-12-01908-f005:**
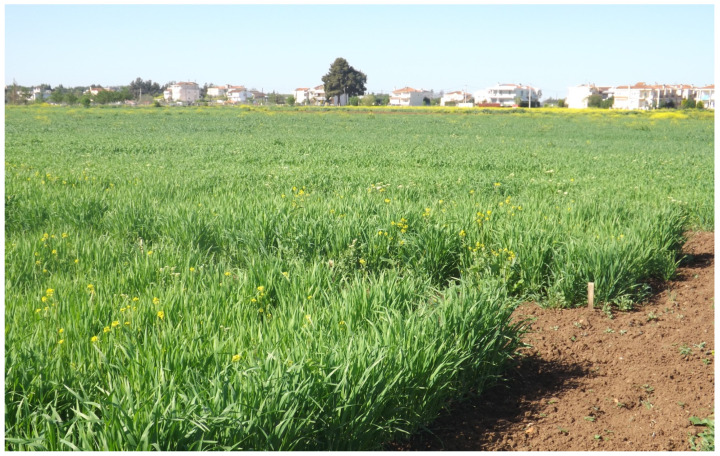
Overview of the two-rowed barley experimental field on 22 April 2016 (90 days after sowing).

**Figure 6 plants-12-01908-f006:**
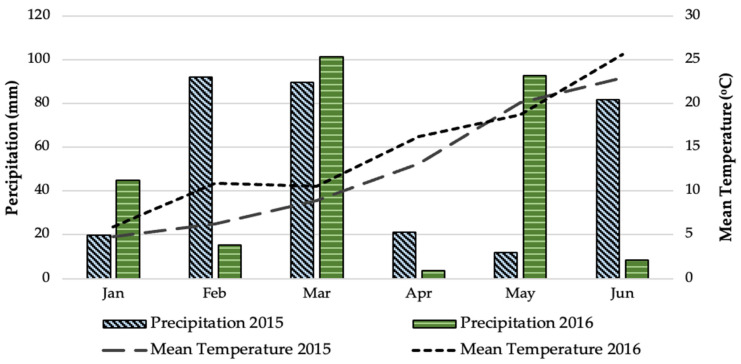
Weather data for the experimental site (Larissa, Thessaly, Greece) during the growing periods (January–June, 2015 and 2016).

**Table 1 plants-12-01908-t001:** Combined analysis of variance (*F* values) for all measured properties of two-rowed barley crop grown under different AMF inoculation and fertilization during the two-year experiment.

**Source of Variance**	**Df**	**Plant Height**	**Leaf Area Index (LAI)**	**Biomass Yield**	**Seed Yield**	**Harvest Index (HI)**	**Biomass Nitrogen (N) Uptake**
Year (Y)	1	90.039 ***	1.371 ^ns^	7.757 **	20.061 ***	0.006 ^ns^	0.088 ^ns^
Inoculation (I)	1	117.121 ***	12.974 ***	35.021 ***	52.184 ***	2.888 ^ns^	6.079 *
Fertilization (F)	4	157.668 ***	14.406 ***	59.011 ***	114.229 ***	1.606 ^ns^	9.976 ***
Y × I	1	0.024 ^ns^	0.027 ^ns^	0.009 ^ns^	0.010 ^ns^	0.001 ^ns^	0.007 ^ns^
Y × F	4	0.052 ^ns^	0.004 ^ns^	0.017 ^ns^	0.033 ^ns^	0.001 ^ns^	0.443 ^ns^
I × F	4	47.888 ***	19.433 ***	1.735 ^ns^	5.113 **	1.016 ^ns^	2.459 ^ns^
Y × I × F	4	0.015 ^ns^	0.004 ^ns^	0.002 ^ns^	0.020 ^ns^	0.002 ^ns^	0.313 ^ns^
**Source of Variance**	**Df**	**Seed N Uptake**	**Total Plant N Uptake**	**Nitrogen Harvest** **Index (NHI)**	**Biomass Phosphorus (P) Uptake**	**Seed P Uptake**	**Total Plant P Uptake**
Year (Y)	1	3.654 ^ns^	2.619 ^ns^	0.151 ^ns^	1.630 ^ns^	15.651 ***	9.813 **
Inoculation (I)	1	4.893 *	9.108 **	2.395 ^ns^	71.370 ***	19.116 ***	68.393 ***
Fertilization (F)	4	31.069 ***	33.613 ***	2.906 *	19.580 ***	112.141 ***	74.012 ***
Y × I	1	0.001 ^ns^	0.001 ^ns^	0.024 ^ns^	0.271 ^ns^	1.121 ^ns^	0.918 ^ns^
Y × F	4	0.009 ^ns^	0.181 ^ns^	0.498 ^ns^	0.039 ^ns^	0.209 ^ns^	0.156 ^ns^
I × F	4	1.893 ^ns^	2.451 ^ns^	2.006 ^ns^	1.259 ^ns^	7.279 ***	2.671 *
Y × I × F	4	0.004 ^ns^	0.106 ^ns^	0.356 ^ns^	0.053 ^ns^	0.180 ^ns^	0.022 ^ns^
**Source of Variance**	**Df**	**Phosphorus Harvest** **Index (PHI)**	**Nitrogen** **Utilization Efficiency (NUtE)**		**Phosphorus Utilization Efficiency (PUtE)**	**AMF** **Colonization**	**Weighted Mycorrhizal Dependency (WMD)**
Year (Y)	1	0.010 ^ns^	0.048 ^ns^		0.042 ^ns^	0.658 ^ns^	0.322 ^ns^
Inoculation (I)	1	52.509 ***	0.008 ^ns^		14.564 ***	63.434 ***	-
Fertilization (F)	4	7.628 ***	3.662 *		6.479 ***	21.626 ***	7.229 ***
Y × I	1	0.006 ^ns^	0.009 ^ns^		0.848 ^ns^	0.434 ^ns^	-
Y × F	4	0.014 ^ns^	0.172 ^ns^		0.099 ^ns^	0.510 ^ns^	0.472 ^ns^
I × F	4	1.783 ^ns^	3.486 *		8.657 ***	4.426 **	-
Y × I × F	4	0.174 ^ns^	0.117 ^ns^		0.011 ^ns^	0.584 ^ns^	-

F-test ratios originated from ANOVA. ns: Non-significant; *, ** and ***: Significant at the 5%, 1% and 0.1% levels, respectively. Df: Degrees of freedom.

**Table 2 plants-12-01908-t002:** Growth and yield parameters of two-rowed barley as affected by AMF inoculation and fertilization treatment.

	Inoculation	Fertilization											
		2015						2016					
		100% Org.	60% Org.	100% Inorg.	60% Inorg.	Control	*Mean*	100% Org.	60% Org.	100% Inorg.	60% Inorg.	Control	*Mean*
**Plant Height** **(cm)**	AMF+	84.2 ^a^	79.1 ^b^	84.8 ^a^	79.0 ^b^	77.7 ^b^	80.9 ^A^	87.2 ^a^	81.9 ^b^	87.8 ^a^	81.9 ^b^	80.5 ^b^	83.9 ^A^
	AMF−	77.5 ^c^	80.1 ^b^	84.1 ^a^	78.8 ^bc^	68.3 ^d^	77.8 ^B^	80.3 ^c^	82.9 ^b^	87.1 ^a^	81.7 ^bc^	70.8 ^d^	80.6 ^B^
	LSD†	***	*	ns	ns	***		***	ns	ns	ns	***	
**LAI** **(m^2^ m^−2^)**	AMF+	4.39 ^a^	3.35 ^b^	3.54 ^ab^	3.22 ^b^	3.17 ^b^	3.53 ^A^	4.52 ^a^	3.45 ^b^	3.63 ^ab^	3.31 ^b^	3.25 ^b^	3.63 ^A^
	AMF−	2.95 ^b^	3.15 ^b^	4.04 ^a^	3.33 ^b^	2.91 ^b^	3.28 ^B^	3.02 ^b^	3.22 ^b^	4.14 ^a^	3.40 ^b^	2.98 ^b^	3.35 ^B^
	LSD†	**	ns	ns	*	ns		**	ns	ns	*	ns	
**Biomass Yield** **(tn ha^−1^)**	AMF+	7.66 ^a^	7.35 ^a^	7.65 ^a^	6.92 ^ab^	6.12 ^b^	7.14 ^A^	7.95 ^a^	7.61 ^a^	7.92 ^a^	7.20 ^a^	6.34 ^b^	7.40 ^A^
	AMF−	7.38 ^a^	6.86 ^ab^	7.34 ^a^	6.25 ^bc^	5.20 ^c^	6.61 ^B^	7.66 ^a^	7.11 ^ab^	7.61 ^a^	6.48 ^b^	5.41 ^c^	6.85 ^B^
	LSD†	ns	**	ns	ns	**		ns	**	ns	ns	*	
**Seed Yield** **(tn ha^−1^)**	AMF+	4.35 ^a^	4.12 ^a^	4.33 ^a^	4.01 ^a^	3.54 ^b^	4.07 ^A^	4.52 ^a^	4.26 ^ab^	4.48 ^ab^	4.18 ^b^	3.67 ^c^	4.22 ^A^
	AMF−	4.25 ^a^	4.05 ^a^	4.12 ^a^	3.62 ^b^	3.11 ^c^	3.83 ^B^	4.41 ^a^	4.21 ^a^	4.27 ^a^	3.75 ^b^	3.24 ^c^	3.98 ^B^
	LSD†	*	ns	*	*	**		*	ns	*	*	**	
**Harvest Index** **(%)**	AMF+	56.82 ^a^	56.03 ^a^	56.62 ^a^	58.04 ^a^	57.89 ^a^	57.08 ^A^	56.91 ^a^	56.00 ^a^	56.63 ^a^	58.19 ^a^	57.88 ^a^	57.12 ^A^
	AMF−	57.69 ^a^	59.10 ^a^	56.20 ^a^	58.09 ^a^	59.72 ^a^	58.16 ^A^	57.75 ^a^	59.15 ^a^	56.21 ^a^	58.10 ^a^	59.89 ^a^	58.22 ^A^
	LSD†	ns	ns	ns	ns	ns		ns	ns	ns	ns	ns	

F-test ratios originated from ANOVA. ns: Non-significant; *, ** and ***: Significant at the 5%, 1% and 0.1% levels, respectively. The different upper- and lower-case letters denote significant differences, under different AMF inoculation treatments and between means of fertilization treatments under the same AMF inoculation treatment, respectively, according to Tukey’s HSD method (*p* < 0.05), and LSD† (*p* < 0.05) denotes significant differences between AMF treatments within the same fertilization treatment. LAI: leaf area index. AMF inoculation treatments: AMF+ and AMF−, with and without AMF inoculation, respectively. Fertilization treatments: 100% and 60% recommended dose of fertilizer in organic form (100% Org and 60% Org, respectively), 100% and 60% recommended dose of fertilizer in inorganic form (100% Inorg. and 60% Inorg, respectively), and untreated (control).

**Table 3 plants-12-01908-t003:** Nitrogen (N) and phosphorus (P) uptake parameters as affected by AMF inoculation and fertilization treatment.

	Inoculation	Fertilization											
		2015						2016					
		100% Org.	60% Org.	100% Inorg.	60% Inorg.	Control	*Mean*	100% Org.	60% Org.	100% Inorg.	60% Inorg.	Control	*Mean*
**Biomass N Uptake** **(kg N ha^−1^)**	AMF+	23.79 ^a^	15.84 ^ab^	18.69 ^ab^	12.02 ^b^	15.25 ^ab^	17.12 ^A^	25.16 ^a^	16.75 ^bc^	19.52 ^ab^	14.19 ^bc^	10.99 ^c^	17.32 ^A^
	AMF−	15.81 ^a^	15.86 ^a^	16.77 ^a^	13.42 ^a^	11.64 ^a^	14.70 ^B^	17.08 ^a^	16.17 ^ab^	17.14 ^a^	13.39 ^ab^	11.54 ^b^	15.06 ^B^
	LSD†	*	ns	*	ns	ns		*	ns	*	ns	ns	
**Seed N Uptake** **(kg N ha^−1^)**	AMF+	76.85 ^a^	68.52 ^ab^	74.15 ^a^	67.36 ^ab^	58.60 ^b^	69.09 ^A^	79.91 ^a^	70.87 ^ab^	76.82 ^a^	70.23 ^ab^	60.68 ^b^	71.70 ^A^
	AMF−	71.28 ^a^	69.05 ^a^	75.56 ^a^	64.61 ^a^	50.33 ^b^	66.17 ^B^	74.00 ^a^	71.69 ^a^	78.34 ^a^	66.92 ^a^	52.51 ^b^	68.69 ^B^
	LSD†	ns	ns	ns	ns	*		ns	ns	ns	ns	*	
**Total Plant N Uptake** **(kg N ha^−1^)**	AMF+	100.64 ^a^	84.37 ^abc^	92.85 ^ab^	79.39 ^bc^	73.85 ^c^	86.22 ^A^	105.07 ^a^	87.62 ^b^	96.34 ^ab^	84.43 ^bc^	71.67 ^c^	89.02 ^A^
	AMF−	87.09 ^a^	84.92 ^a^	92.33 ^a^	78.03 ^ab^	61.97 ^b^	80.87 ^B^	91.08 ^a^	87.87 ^a^	95.48 ^a^	80.31 ^ab^	64.05 ^b^	83.76 ^B^
	LSD†	*	ns	ns	ns	*		*	ns	ns	ns	*	
**NHI (%)**	AMF+	76.37 ^b^	81.17 ^ab^	79.97 ^ab^	84.80 ^a^	79.93 ^ab^	80.45 ^A^	76.00 ^b^	80.83 ^ab^	79.87 ^ab^	83.20 ^ab^	84.67 ^a^	80.91 ^A^
	AMF−	82.10 ^a^	81.30 ^a^	82.10 ^a^	82.75 ^a^	81.27 ^a^	81.91 ^A^	81.57 ^a^	81.60 ^a^	82.10 ^a^	83.30 ^a^	81.97 ^a^	82.11 ^A^
	LSD†	ns	ns	ns	ns	ns		ns	ns	ns	ns	**	
**Biomass P Uptake** **(kg P ha^−1^)**	AMF+	4.20 ^ab^	4.67 ^a^	3.94 ^ab^	3.58 ^bc^	2.94 ^c^	3.87 ^A^	4.46 ^ab^	4.83 ^a^	4.15 ^abc^	3.70 ^bc^	3.16 ^c^	4.06 ^A^
	AMF−	3.19 ^ab^	3.41 ^a^	3.11 ^ab^	2.96 ^ab^	2.40 ^b^	3.01 ^B^	3.29 ^ab^	3.55 ^a^	3.26 ^ab^	3.05 ^ab^	2.33 ^b^	3.09 ^B^
	LSD†	*	**	*	ns	ns		*	**	*	ns	*	
**Seed P Uptake** **(kg P ha^−1^)**	AMF+	9.63 ^a^	8.16 ^b^	9.50 ^a^	7.85 ^b^	6.87 ^c^	8.40 ^A^	10.16 ^a^	8.66 ^b^	10.07 ^a^	8.33 ^b^	7.05 ^c^	8.85 ^A^
	AMF−	8.62 ^a^	8.53 ^ab^	9.04 ^a^	7.59 ^bc^	6.75 ^c^	8.11 ^B^	8.90 ^a^	8.88 ^a^	9.32 ^a^	7.71 ^b^	7.01 ^b^	8.37 ^B^
	LSD†	***	ns	ns	ns	ns		***	ns	*	ns	ns	
**Total Plant P Uptake** **(kg P ha^−1^)**	AMF+	13.84 ^a^	12.83 ^a^	13.43 ^a^	11.43 ^b^	9.81 ^c^	12.27 ^A^	14.62 ^a^	13.49 ^a^	14.21 ^a^	12.03 ^b^	10.21 ^c^	12.91 ^A^
	AMF−	11.80 ^a^	11.94 ^a^	12.15 ^a^	10.55 ^ab^	9.15 ^b^	11.12 ^B^	12.20 ^a^	12.43 ^a^	12.58 ^a^	10.76 ^ab^	9.34 ^b^	11.46 ^B^
	LSD†	**	*	*	ns	ns		**	*	*	ns	*	
**PHI (%)**	AMF+	69.67 ^a^	63.60 ^b^	70.77 ^ab^	68.77 ^ab^	70.03 ^a^	68.57 ^B^	69.50 ^a^	64.10 ^b^	70.86 ^a^	69.26 ^a^	69.15 ^a^	68.58 ^B^
	AMF−	73.13 ^a^	71.47 ^a^	73.87 ^a^	72.27 ^a^	74.43 ^a^	73.03 ^A^	73.07 ^a^	71.46 ^a^	74.13 ^a^	71.97 ^a^	75.10 ^a^	73.15 ^A^
	LSD†	ns	*	*	ns	ns		ns	*	*	ns	*	

F-test ratios originated from ANOVA. ns: Non-significant; *, ** and ***: Significant at the 5%, 1% and 0.1% levels, respectively. The different upper- and lower-case letters denote significant differences, under different AMF inoculation treatments and between means of fertilization treatments under the same AMF inoculation treatment, respectively, according to Tukey’s HSD method (*p* < 0.05), and LSD† (*p* < 0.05) denotes significant differences between AMF treatments within the same fertilization treatment. NHI: nitrogen harvest index, PHI: phosphorus harvest index. AMF inoculation treatments: AMF+ and AMF−, with and without AMF inoculation, respectively. Fertilization treatments: 100% and 60% recommended dose of fertilizer in organic form (100% Org and 60% Org, respectively), 100% and 60% recommended dose of fertilizer in inorganic form (100% Inorg. and 60% Inorg, respectively), and untreated (control).

## Data Availability

Not applicable.

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
