# Peer review of "Effect of Arbuscular Mycorrhizal Fungi on Nitrogen and Phosphorus Uptake Efficiency and Crop Productivity of Two-Rowed Barley under Different Crop Production Systems"

_plants, 2023, doi:10.3390/plants12091908_

Round 1
Reviewer 1 Report
This research is interesting and informative. Before being published, it needs to be revised in the following points.
1. I am not sure if the current structure is consistent with the requirements of the Journal or not. If not, I suggest moving the part of the study area and methods to part 2 following the introduction.
2. I also suggest adding a description of the local soil Microorganisms, at least AMF types.
3. Rename the different treatments (10), which can help understand the figures' content.
4. The results only show the measured data, which is insufficient and hinders the deep understanding of the influence of different treatments (especially interactive) on plants. I suggest calculating the difference between different treatments. For example, comparing the plant factors values of the AMF+ induced minus AMF- induced in the different fertilization treatments, and vice versa. The authors can find more informative results.
5. The discussion part is also not satisfactory. Most is the repetition of results. The mechanism analysis is lacking.
6. I also suggest the authors analyzing year-induced significant differences.
Author Response
Reviewer 1
This research is interesting and informative. Before being published, it needs to be revised in the following points.
Response: We thank the reviewer for the positive comments on our study and for the interesting information useful for improving it.
- I am not sure if the current structure is consistent with the requirements of the Journal or not. If not, I suggest moving the part of the study area and methods to part 2 following the introduction.
Response: The current structure of manuscript is consistent with the requirements of the Journal. Please see Research Manuscript Sections on Instructions for Authors section (https://www.mdpi.com/journal/plants/instructions)
- I also suggest adding a description of the local soil Microorganisms, at least AMF types.
Response: We revised according to the reviewer’s request. The changes were marked up using the “Track Changes” function. (“The genera Glomus and Gigaspora have been reported to be the dominant AM fungi in the arable lands of Greece [89,90].”) (Lines 661-662).
- Rename the different treatments (10), which can help understand the figures' content.
Response: We revised according to the reviewer’s request. Specifically, on each table or figure, we added a note that explains the abbreviation of each treatment. The changes were marked up using the “Track Changes” function. (For instance, Figure 3. Weighted mycorrhizal dependency (WMD) of two-rowed barley crop as affected by different fertilization type and level. Error bars represent + S.E. Different letters above bars represent significant differences at p < 0.05 according to Tukey’s HSD test. Fertilization treatments: 100% and 60% recommended dose of fertilizer in organic form (100% Org and 60% Org, respectively), 100% and 60% recommended dose of fertilizer in inorganic form (100% Inorg. and 60% Inorg, respectively), and untreated (control)).
- The results only show the measured data, which is insufficient and hinders the deep understanding of the influence of different treatments (especially interactive) on plants. I suggest calculating the difference between different treatments. For example, comparing the plant factors values of the AMF+ induced minus AMF- induced in the different fertilization treatments, and vice versa. The authors can find more informative results.
Response: Although the suggested approach is interested, we believe that in the current form the text of the results section clearly explains the results of our study. The influence of different treatments is both presented and discussed and in addition, in the current form, we avoid overlapping information.
- The discussion part is also not satisfactory. Most is the repetition of results. The mechanism analysis is lacking.
Response: We revised this part according to the reviewer’s request. We believe that in the current form the text of the discussion section clearly explains and discusses the results of our study. In addition, we added a paragraph on mechanism analysis which was missing from the text.
- I also suggest the authors analyzing year-induced significant differences.
Response: We believe that in the current form the text of the results section clearly explains the results of the current study. Moreover, we avoid overlapping information.
Reviewer 2 Report
This study was well-designed, well-conducted, and well-presented. It was a pleasure to review a manuscript of this quality. I learned a lot about how to present and interpret 3-way ANOVA from reading this study. The English language in this manuscript is fluent and error-free. Although this study was conducted 8 years ago, its results are still valuable today, as biostimulants are becoming increasingly popular.
I have one question: were the two main plots (AMF+ and AMF-) located in the same spot in both 2015 and 2016? The inoculation of AMF is a treatment, and the inoculated AMF in 2015 may have affected the soil condition in 2016, which could explain the difference in phosphorus uptake between the two harvest years.
Author Response
Reviewer 2
This study was well-designed, well-conducted, and well-presented. It was a pleasure to review a manuscript of this quality. I learned a lot about how to present and interpret 3-way ANOVA from reading this study. The English language in this manuscript is fluent and error-free. Although this study was conducted 8 years ago, its results are still valuable today, as biostimulants are becoming increasingly popular.
Response: We thank the reviewer for the positive comments on our study and for the interesting information useful for improving it.
I have one question: were the two main plots (AMF+ and AMF-) located in the same spot in both 2015 and 2016? The inoculation of AMF is a treatment, and the inoculated AMF in 2015 may have affected the soil condition in 2016, which could explain the difference in phosphorus uptake between the two harvest years.
Response: We thank the reviewer for the well pointed out remark. Both main plots and sub plots were relocated in the same field during second year and inoculation was repeated. We added a clarification in the text (Lines 585-586).
Reviewer 3 Report
The present article entitled "Effect of Arbuscular Mycorrhizal Fungi on Nitrogen and Phosphorus Uptake Efficiency and Crop Productivity of Two-rowed Barley under Different Crop Production Systems" cover a very interesting topic of AMF and their inoculation effect. The experiment is well designed and the presentation of the article is well. I recommend acceptance of article in the present form.
The quality of english and presentation is well
Author Response
Reviewer 3
The present article entitled "Effect of Arbuscular Mycorrhizal Fungi on Nitrogen and Phosphorus Uptake Efficiency and Crop Productivity of Two-rowed Barley under Different Crop Production Systems" cover a very interesting topic of AMF and their inoculation effect. The experiment is well designed and the presentation of the article is well. I recommend acceptance of article in the present form.
Response: We thank the reviewer for the positive comments on our study.